# Chemical Structures and Antioxidant Activities of Polysaccharides from *Carthamus tinctorius* L.

**DOI:** 10.3390/polym14173510

**Published:** 2022-08-26

**Authors:** Dan Lin, Cheng-Jian Xu, Yang Liu, Yu Zhou, Shuang-Li Xiong, Hua-Chang Wu, Jing Deng, Yu-Wen Yi, Ming-Feng Qiao, Hang Xiao, Sook-Wah Chan, Yi Lu

**Affiliations:** 1Department of Food Science, College of Food Science and Technology, Sichuan Tourism University, Chengdu 610100, China; 2Faculty of Health and Medical Sciences, Taylor’s University, Subang Jaya 47500, Selangor Darul Ehsan, Malaysia; 3Key Laboratory of Xinjiang Phytomedicine Resource and Utilization, Ministry, College of Food Sciences, Shihezi University, Shihezi 832000, China; 4Department of Food Science, University of Massachusetts, Amherst, MA 01003, USA or

**Keywords:** *Carthamus tinctorius* L., polysaccharide, chemical structure, antioxidant activity

## Abstract

Two polysaccharides from *Carthamus tinctorius* L. (CTLP-1 and CTLP-2) were purified, and their structures were analyzed by physical and chemical testing. CTLP-1 had a mass of 5900 Da that was composed of arabinose, glucose, and galactose with a mass molar ratio of 6.7:4.2:1. The backbone of CTLP-1 was →1)-α-GalAp-(1→4)-α-Arap-(1→2)-α-Glup-(4→. CTLP-2 had a mass of 8200 Da that was composed of arabinose, glucose, and galactose with a mass molar ratio of 16.76:4.28:1. The backbone of CTLP-2 was →1)-α-Galp-(2,6 →1)-α-Arap-(4,6 →1)-α-Glup-(3→. Both of them exhibited a high reducing power, hydroxyl radical scavenging activity, DPPH radical scavenging activity and ABTS radical scavenging activity, moderate Fe^2+^ chelating activity and superoxide anion scavenging activity, implying that they might be potential antioxidants.

## 1. Introduction

Safflower (*Carthamus tinctorius* L.) is an annual herb native to the sandy lands of Russia, Japan, Korea, and China [1]. Among the planting areas in China, Xinjiang has the best light conditions and is the most suitable for safflower growth [2]. In 1998, Tacheng, Xinjiang was officially named the “hometown of Chinese safflower” by the Ministry of Agriculture. Safflower has been used in traditional medicine for thousands of years to remove blood stasis and activate blood circulation [1]. Modern pharmacological research has shown that the long-term intake of safflower is good for improving immunity [3] and reducing incidences of hypertension [4] and cancer [5]. As a result of the edible and medicinal value of safflower, people’s interest in safflower products is growing day-by-day. Many safflower products are currently on the market, including painkillers, health drinks, skin lotions, tablets, and other nutritional supplements.

A range of bioactive molecules has been extracted from *C. tinctorius* L., including carthamin, precarthamin, safflower yellow, safflomin A, and some polyphenols [6,7]. *C*. *tinctorius* L. also contains a considerable amount of soluble polysaccharides that show immunomodulatory [8], anti-diabetic [9], anti-cancer [10], and anti-tumor [11] activities. These polysaccharides can be extracted under certain conditions to reduce food waste and improve the added value of agricultural products [12]. In a previous study, our laboratory optimized the extraction conditions of polysaccharides from *C. tinctorius* L. (CTLP), and high CTLP yield was obtained in our published study [13]. However, little is known about the structure and antioxidant activity of safflower polysaccharides. At present, in China, especially in Xinjiang, where safflower is highly produced, safflower is commonly used as a general food or Chinese herbal medicine, but its activity has not been definitely clarified so far. The study of its antioxidant activity will help analyze its physiological activity and provide a basis for its deep processing industry.

In this study, the structures and antioxidant activities of CTLP were investigated. Soluble polysaccharides were extracted and purified from the *C. tinctorius* L. The structures of the CTLP were analyzed by high-performance liquid chromatography (HPLC), high-pressure exclusion chromatography (HPSEC), scanning electron microscopy (SEM), Fourier transform infrared (FT-IR) spectroscopy, and nuclear magnetic resonance (NMR) spectroscopy. The antioxidant activities were evaluated based on the reducing power, Fe^2+^ chelation ability, superoxide, hydroxyl, 2,2-diphenyl-1-picrylhydrazyl (DPPH), and 2,2’-azino-bis(3-ethylbenzothiazoline-6-sulfonic acid) (ATBS) radical scavenging activities. The molecular weight, monosaccharide composition and main sugar chain skeleton of CTLP were determined. CTLP exhibited a high reducing power, hydroxyl radical scavenging activity, DPPH radical scavenging activity and ABTS radical scavenging activity, moderate Fe^2+^ chelating activity and superoxide anion scavenging activity.

## 2. Materials and Methods

### 2.1. Experimental materials

*C. tinctorius* L. was kindly provided by Xinjiang Academy of Agricultural Sciences (Xinjiang, China), purchased from local herdsmen in Tacheng, Xinjiang, China. DEAE-52 resin, Sephadex G-100 medium and dextran standards were ordered from Sigma-Aldrich (St. Louis, Missouri, USA). All chemicals and reagents mentioned in this study were analytical or chromatographic grade.

### 2.2. Purification of CTLP

CTLP was extracted in the laboratory under optimized conditions. According to the method of Yang [10], the DEAE-52 cellulose column material was pre-treated for activation, and the activated column material was dissolved in distilled water. The column was slowly loaded, and the appropriate pump pressure was selected for column balance. After the column material was completely settled, 15 mg of CTLP was precisely weighed and dissolved in a 10 mL test tube using distilled water. After centrifugation for 20 min, the supernatant solution was sampled. The pump pressure was controlled between 3.0 bar and 3.5 bar using a peristaltic pump, and the flow rate was maintained between 0.8–1.0 mL∙min^−1^. The CTLP adsorbed on the cellulose column was eluted with water along with different concentrations of NaCl solution (0.1, 0.2, 0.4, and 0.8 mol∙L^−1^). The collection time of the automatic collector was 8 min. The polysaccharide content curve was recorded for each polysaccharide solution. According to the curves, the same component CTLP was collected, freeze-dried, and saved for purification.

The Sephadex G-100 material was soaked in water for 24 h in a refrigerator at 4 °C, rinsed with NaOH solution (0.5 mol∙L^−1^) for 0.5 h, and then rinsed repeatedly with water until a neutral pH was obtained. The pure polysaccharides were obtained by freeze-drying the collected eluent based on the elution curve, and the purity of each polysaccharide was verified by their absorbance values.

The polysaccharide content was determined at 470 nm with an enzyme mark instrument (ELx808, Hercules California, USA) following the DNS (dinitrosalicylic acid) method [14] with glucose as the standard.

### 2.3. Molecular Weight Detection of CTLP

The molecular weight of CTLP was detected by HPSEC [15]. After column balancing, T-10, T-40, T-70, and T-500 glucan solutions were successively prepared. The concentration of the dextran solution was 3 mg∙mL^−1^, the pump pressure was 3.5 bar, and the flow rate was 2.5 mL∙min^−1^. The dextran solution was loaded into the column, and the dextran solution was fully eluted with NaCl solution (0.1 mol∙L^−1^), respectively. The elution volume of each glucan solution was recorded as *V_e_*. Blue dextran 2000 was treated in the same way, and the volume of blue glucan 2000 eluent was recorded as *V_0_*, where *V_t_* is the volume of Sephadex G-100. The *K* values of the different glucan solutions were then calculated using Equation (1).
(1)$$K=\frac{V_{e}-V_{0}}{V_{t}-V_{0}}$$

The *K* value was set as the ordinate and the log [*M*] curve was then drawn as the standard curve as the abscissa, where *M* is the standard molecular weight of glucan. The molecular weight of CTLP was determined in the same way.

### 2.4. Monosaccharide Composition Analysis

CTLP solution was loaded onto a reverse-phase C18 column (4.6 × 25 cm) to achieve chromatographic separation. The injection volume was 20 μL, and the column temperature was 37 °C. Pure acetonitrile (mobile phase A) was passed through the organic membrane (0.22 μm). Mobile phase B was formed by dissolving 0.45 g KH_2_PO_4_ in 100 mL of acetonitrile, adding 0.5 mL of TEA and 900 mL of ultra-pure water, shaking well, and passing it through the membrane. The column was balanced with phase A for 3 min, and the elution procedure was as follows: 94% mobile phase B (0–4 min); 94–88% mobile phase B (4–9 min); 88% mobile phase B (9–30 min); and 88–94% mobile phase B (30–35 min).

### 2.5. Periodate Oxidation

NaIO4 solutions (30 nmol∙L^−1^; 0, 0.5, 1.0, 1.5, 2.0, and 4.0 mL) were diluted 250 times with purified water, and their absorbance values at 223 nm were determined [16]. The standard curve of periodate was obtained by taking the concentration as the abscissa and the absorbance as the ordinate. CTLP (25 mg) was fully dissolved in 15 mL of NaIO_4_ solution (50 nmol∙L^−1^) and stored in a brown volumetric bottle to keep light out. The absorbance at 223 nm was repeatedly measured until the final absorbance value remained unchanged. The mixture with constant absorbance (2 mL) was titrated with 0.01 mol∙L^−1^ NaOH solution, and phenolphthalein was used as an indicator to determine the production of formic acid. Using purified water as the control, the consumption of periodate was calculated with the standard curve.

### 2.6. FT-IR Analysis of CTLP

The purified CTLP was ground with KBr powder. The crude CTLP was verified in the range of 4000–400 cm^−1^ by FT-IR spectroscopy (IRPrestige-21, Shimadzu, Japan) following the KBr pressed-disk method [17,18]. Each sample was detected three times.

### 2.7. NMR Spectroscopy

The CTLP was dissolved in D_2_O at a concentration of 3% for 3 h and then freeze-dried. The ^1^H spectra of CTLP was detected with an NMR spectrometer (500 MHz, ^1^H; VNMRS600, Agilent).

### 2.8. Congo Red Analysis

To determine the triple-helical structures of the CTLP, refer to the method of our previous research [19]. The absorbance was recorded at 470 nm with different concentrations of Congo red solution and compared with the absorbance of distilled water as a control.

### 2.9. SEM Anaylsis of CTLP

For SEM analysis [20] of CTLP, the purified CTLP was fixed in 3% glutaraldehyde (0.1 M sodium cacodylate pH 7.4, 5 mM CaCl_2_) overnight at 23 °C, then stained for 1 h at 23 °C in 1% osmium tetroxide and 1% potassium ferricyanide in cacodylate buffer, washed with ddH_2_O and then dehydrated by 10 min incubations in a graded acetone series: two incubations each in 30%, 50% and 70% acetone, and four incubations in 90% and 100% acetone. Then the cover slips were critical-point dried and sputter-coated with 4 nm of gold particles. Images were collected in an LEO1550 scanning electron microscope at 2.5 kV using an in-lens detector.

### 2.10. Antioxidant Activity Assays

The reducing power of the CTLP was determined by the following method [21]: The solutions of CTLP with different concentrations were added into different test tubes with the addition of a 1 mL phosphate buffer (1% potassium ferricyanide, pH 6.6), made up to 4.5 mL with water. The test tube was heated in a water bath for 20 min at 50 °C with the addition of 2.5 mL 10% trichloroacetic acid solution. The supernatant was obtained by centrifugation for 20 min at 8000× *g*. The supernatant was transferred to a test tube with 0.5 mL 0.1% FeCl_3_ solution and 2.5 mL water. The absorbance of the mixture was determined at 700 nm, and vitamin C (*Vit. C*) was the positive control.

The Fe^2+^ chelating activity of the CTLP was determined as follows [22]: Five portions of EDTA solution or CTLP solution with concentration gradient were added into 0.05 mL 2 mmol∙L^−1^ ferrous chloride solution, 0.2 mL 5 mmol∙L^−1^ Ferrozine solution and 2.75 mL water. After fully mixing, the absorbance was determined as 562 nm. EDTA was the positive control, and the water was the blank control.

The superoxide anion radical scavenging activity was determined by a previous method [23]. Different concentrations of 0.5 mL CTLP solution, 0.5 mL, 25 mmol∙L^−1^ pyrogallol; and 2 mL, 50 mmol∙L^−1^ Tris-HCl were mixed. Then, the mixture was incubated at 25 °C for 5 min with the addition of hydrochloric acid, and the reaction was stopped. The absorbance was recorded at 560 nm with *Vit. C* as the positive control.

The hydroxyl radical scavenging activity was determined as previously reported [24]. The 1 mL, 9 mmol∙L^−1^ CTLP was mixed 1 mL, 9 mmol∙L^−1^ salicylic acid-ethanol and 1 mL, 9 mmol∙L^−1^ FeSO_4_. The mixture was incubated at 37 °C for 30 min. The absorbance was detected at 510 nm with *Vit. C* as the positive control.

The DPPH radical scavenging activity of the CTLP was detected as follows [22]. 2 mL different concentrations CTLP with 2.5 mL DPPH were mixed, respectively. The mixture was incubated at 25 °C for 30 min. The absorbance was detected at 517 nm, and *Vit. C* was the positive control.

The ABTS radical scavenging activity was detected, as in the previous description [25]. Different concentrations of CTLP were mixed with 3 mL ABTS solution, then incubated at 25 °C for 6 min. The absorbance detected at 510 nm was recorded and the (*Vit. C*) was again used as the control group.

The scavenging activities above were determined as Equation (2),
(2)$$\left(1-\frac{A_{1}-A_{2}}{A_{0}}\right)\times 100$$
where *A*_0_ is the absorbance of doubly distilled water, *A*_1_ is the absorbance of the solution with CTLP in the reactive system, and *A_2_* is the absorbance of the solution with CTLP.

## 3. Results and Discussion

### 3.1. Structural Analysis

The elution curve of safflower polysaccharides adsorbed on the cellulose column was obtained by gradient elution with water and NaCl solutions with gradient concentrations. Two uniform elution peaks are seen in Figure 1A. According to the elution curve, two CTLP components were obtained and marked as CTLP-1 and CTLP-2; CTLP-1 was eluted by 0.1 mol∙L^−1^ NaCl solution and CTLP-2 was eluted with 0.2 mol∙L^−1^ NaCl solution. The absorbance was determined by the DNS method. The CTLP-1 and CTLP-2 solutions were dialyzed in double-steaming water by dialysis bags with an interception molecular weight of 4000 Da at 4 °C for 24 h, and the polysaccharides were collected by freeze-drying the dialysate.

The collected CTLP-1 and CTLP-2 were eluted using a G-100 agarose gel column, and the absorbance of each eluent was determined. As shown in Figure 1A,B, the elution curves of CTLP-1 and CTLP-2 both show single symmetrical peaks, indicating a high purity [17]. Thus, the employed separation method was effective for the separation of safflower polysaccharides. The ultraviolet spectra of CTLP-1 and CTLP-2 (Figure 2) were measured in the region of 200–400 nm. No obvious absorption peaks were observed in the ultraviolet spectra. The spectra of CTLP-1 and CTLP-2 were smooth at 260 nm and 280 nm, which means that the polysaccharide did not contain nucleic acid.

The standard curve of glucan molecular weight was obtained and the regression equation was *Y* = 5.9207-5.9096*X* (*R*^2^ = 0.9984). The relative molecular weights of CTLP-1 and CTLP-2 were 5900 and 8200 Da, respectively. The molecular weights of plant polysaccharides have been reported to affect their biological activities to a certain extent. In general, an overly small molecular weight corresponds to a weak biological activity, whereas a high molecular weight leads to strong activity. Compared with the leaf polysaccharide, the root polysaccharide of *Panax ginseng* marc was more complex and had a larger molecular weight. The immune activity of the root polysaccharide was higher than that of the leaf polysaccharide [18]. Similarly, three polysaccharides were isolated and purified from potato peel. The polysaccharide with a larger molecular weight showed a higher antioxidant activity in vitro [22]. However, this does not mean that the activity is proportional to the molecular weight. Compared to similar plant polysaccharides, the molecular weights of safflower polysaccharides are intermediate.

The common monosaccharides in plant polysaccharides are galactose, glucose, arabinose, fructose, xylose, rhamnose, and various other monosaccharides. Figure 3A shows the HPLC chromatogram of the monosaccharide standard. CTLP-1 was a heteropolysaccharide composed of arabinose, glucose, and galactose with a molar ratio of 6.7:4.2:1 (Figure 3B). CTLP-2 was also a heteropolysaccharide containing arabinose, glucose, and galactose with a molar ratio of 16.76:4.28:1 (Figure 3C). Some scholars believe that polysaccharide activity is related to the type and content of polysaccharides. Polysaccharides with a high glucose content are reported to have strong immune activity, while polysaccharides with more kinds of monosaccharides exhibit a high antioxidant activity [18]. However, these trends are not seen in all studies, and specific relationships remain to be determined.

Periodate oxidation primarily occurs on the sugar chains of polysaccharides and selectively acts on the trihydroxyl and hydroxyl groups of the sugar chains. The periodate consumption and formic acid production differ based on the reaction conditions [16]. The standard curve of sodium periodate for the CTLP is given by *Y* = 0.0735*X* + 0.0569 (*R*^2^ = 0.9986). According to the standard curve, CTLP-1 and CTLP-2 were gradually oxidized by periodate, and the absorbance decreased continuously until becoming stable. This process consumed periodate (2.31 and 2.14 mmol for CTLP-1 and CTLP-2, respectively) and produced formic acid (0.92 and 0.84 mmol for CTLP-1 and CTLP-2, respectively). For both CTLP-1 and CTLP-2, the amount of periodate consumed was more than twice that of formic acid produced. Thus, the CTLP had a bond type that consumed periodate but did not produce formic acid (i.e., 1→6 glycosidic bond). The oxidized residue of CTLP-1 was approximately 70%, and the non-oxidizable residue was approximately 30%; thus, CTLP-1 had 1→6 glycosidic bonds and a small amount of 1→3 glycosidic bonds. The oxidized residues of CTLP-2 were approximately 50%. The oxidation residues of 1→2 and 1→4 were approximately 10%, and the non-oxidizable residues were approximately 40%. Thus, CTLP-2 had 1→6 glycosidic bonds and a small amount of 1→2 and 1→4 glycosidic bonds.

The FT-IR spectrum of CTLP-1 (Figure 4A) had obvious absorption peaks at 3383, 2920, 1664, 1379, 1043, and 623 cm^−1^. The peak at 3383 cm^−1^ is attributed to the stretching vibrations of the –OH and N–H functional groups. This peak is not sharp, indicating that these functional groups were in a non-free state [17]. The peak at 2920 cm^−1^ is mainly caused by the stretching vibration of the –CH_2_– functional group. The absorption peak at 1664 cm^−1^ is caused by the stretching vibration of C=O. The peak at 1379 cm^−1^ is caused by the angular vibration of the N–H functional group. The peak at 1043 cm^−1^ is mainly caused by the angular vibration of C–H. Finally, the peak at 623 cm^−1^ is due to the C–O stretching vibration in the C–O–C functional group of the ether bond of the pyran ring [26]. The spectrum of CTLP-2 (Figure 4B) shows obvious absorption peaks at 3502, 2908, 1708, 1527, 1409, and 1062 cm^−1^, indicating a similar functional group composition to CTLP-1. The absorption peaks at 1043 and 1046 cm^−1^ indicate that both CTLP-1 and CTLP-2 contained pyran rings. For CTLP-1, the absorption peak at 623 cm^−1^ indicates the existence of glycosidic bonds with α configuration.

As shown in Figure 5, the ^1^H NMR chemical shifts of CTLP-1 are mainly observed in the region of 0.3–2.7 ppm, while those of CTLP-2 mainly occur in the range of 0.3–4.5 ppm. This indicates that the glycosidic bonds in the CTLP were mainly α-glycosidic bonds. CTLP-1 exhibited fewer NMR peaks than CTLP-2, indicating fewer residues in CTLP-1 [27]. The highest peak in the NMR spectrum of CTLP-1 is the solvent peak of heavy water, and two other peaks are observed. The results indicate that the main chain was composed of right α-D-glucopyranose and galactose. In contrast, CTLP-2 had more peaks in the range of 0.3–4.5 ppm, and thus more residues in the sugar chain. CTLP-2 was mainly composed of α-D-glucopyranose, galactose, and arabinose. 

For the mixture of CTLP-1 and CTLP-2 with Congo red, the wavelength of maximum absorption continuously decreased with the increase in sodium hydroxide concentration, due to the safflower polysaccharides reacting with Congo red to form new complexes [19,28]. As the NaOH concentration increased, the maximum absorption wavelength of CTLP-1 gradually decreased (Figure 6); however, the difference between the CTLP-1 and Congo red absorption curves were small, indicating that CTLP-1 did not have a triple-helix structure. In contrast, the decreasing trend of CTLP-2 was more obvious because the sugar chain of CTLP-2 was broken, and the helix structure was dissociated. As the degree of dissociation increased, the wavelength of maximum absorption continuously decreased until becoming stable; thus, CTLP-2 had a certain helix structure.

The SEM images of CTLP-1 (Figure 7A) and CTLP-2 (Figure 7B) show flake-like, irregular structures. Under 5000× magnification, the particle size of safflower polysaccharide was 10 μm, CTLP-1 was rod-shaped, flake-shaped, and spherical with a non-smooth surface and a small internal gap. The surface of CTLP-2 was smooth with an intermolecular gap, which might be due to the repulsion between the molecules [29] and the distance between the molecules [26].

### 3.2. Antioxidant Activities of CTLP

CTLP-1 and CTLP-2 exhibited a certain degree of reducing power in Figure 8A, and the reducing power and polysaccharide concentration were positively correlated. Both of them exhibited a high reducing power, and the reducing power of CTLP-2 was stronger than that of CTLP-1 when the polysaccharide concentration exceeded 1.8 mg∙mL^−1^. This difference may be related to their respective molecular weights and glycosidic bonds [18]. Compared to CTLP-1, CTLP-2 had a larger molecular weight, more NMR peaks and more functional groups in the free state. Thus, the reducing power of CTLP-2 became stronger than that of CTLP-1.

As shown in Figure 8B, with the polysaccharide concentration of 1.8 mg∙mL^−1^, the Fe^2+^ chelating activities of CTLP-2 and CTLP-1 reached 29.5% and 25.2%, respectively. CTLP-1 and CTLP-2 had weaker Fe^2+^ chelating activities than EDTA which might be related to the fact that Fe^2+^ is easier to chelate in alkaline environments [19], while the water-extracted safflower polysaccharides were neutral.

CTLP-1 and CTLP-2 showed concentration-dependent superoxide anion radical scavenging activities (Figure 8C), but both of them had low activities at 1.8 mg∙mL^−1^. This may be related to the monosaccharide composition of safflower polysaccharides. The monosaccharide content has a certain effect on the antioxidant activity of polysaccharides, and the specific monosaccharide composition can show different antioxidant activity in different antioxidant indexes [22].

We can see that CTLP-1 and CTLP-2 exhibited strong hydroxyl radical scavenging activity compared with *Vit. C* as shown in Figure 8D, and scavenging activity was positively correlated with the concentration of polysaccharide. CTLP-1 (47.2%) showed a stronger hydroxyl radical scavenging activity than CTLP-2 (44.5%) at the concentration of 1.8 mg∙mL^−1^.

In Figure 8E, the DPPH radical scavenging activity of CTLP-2 was similar to that of CTLP-1, consistent with the scavenging activities for hydroxyl radical and superoxide anion radical. The DPPH radical scavenging activities of CTLP-1 and CTLP-2 increased with increasing polysaccharide concentration. The maximum DPPH radical scavenging activities were obtained at the concentration of 1.8 mg∙mL^−1^ (65.2% for CTLP-1 and 60.5% for CTLP-2).

We also noted that the ATBS radical scavenging activities of CTLP-1 and CTLP-2 increased with increasing polysaccharide concentration (Figure 8F). The activities of CTLP-1 and CTLP-2 reached 49.7% and 43.8%, but compared with the activity of *Vit. C*, the ATBS radical scavenging activities of CTLP-1 and CTLP-2 were strong enough.

CTLP1 and CTLP2 exhibited different antioxidant activities in different experiments, and there were some differences between them. The difference between their respective molecular weights and glycosidic bonds may lead to the differentiation. Therefore, the effect of polysaccharide structure on antioxidant activities needs to be further studied.

## 4. Conclusions

The two CTLP components were obtained and marked as CTLP-1 and CTLP-2 by using a G-100 agarose gel column from *Carthamus tinctorius* L. The relative molecular weights of CTLP-1 and CTLP-2 were 5900 and 8200 Da, respectively. CTLP-1 was a heteropolysaccharide composed of arabinose, glucose, and galactose with a molar ratio of 6.7:4.2:1; CTLP-2 was a heteropolysaccharide containing arabinose, glucose, and galactose with a molar ratio of 16.76:4.28:1. The results show that both CTLP-1 and CTLP-2 exhibited relatively high antioxidant activities in vitro and could be further applied as a novel natural antioxidant drug. In the future, we would look to measure the activity of CTLPs in vivo and to research the relationship between its activity and structure.

## Figures and Tables

**Figure 1 polymers-14-03510-f001:**
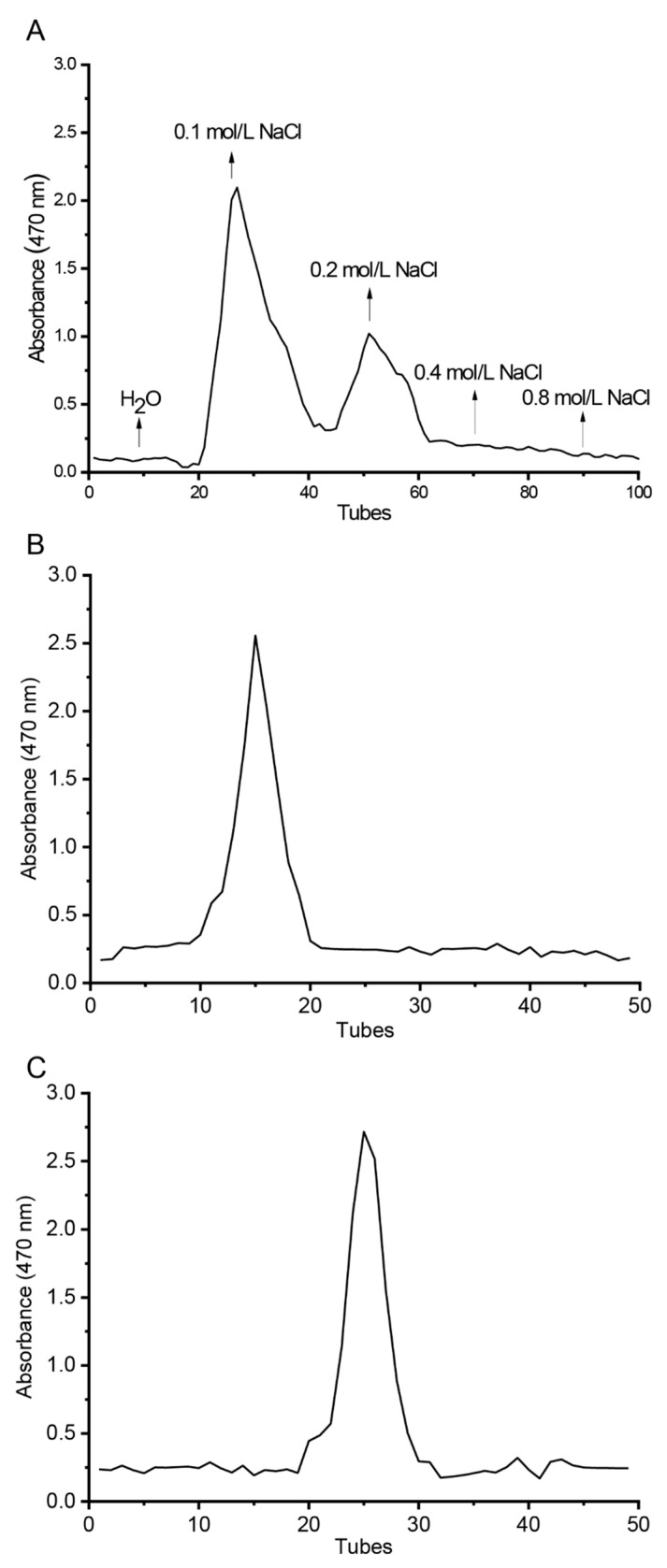
DEAE-52 elution curve of the CTLP (**A**), elution patterns of CTLP-1 (**B**), and CTLP-2 (**C**) on Sephadex G-100.

**Figure 2 polymers-14-03510-f002:**
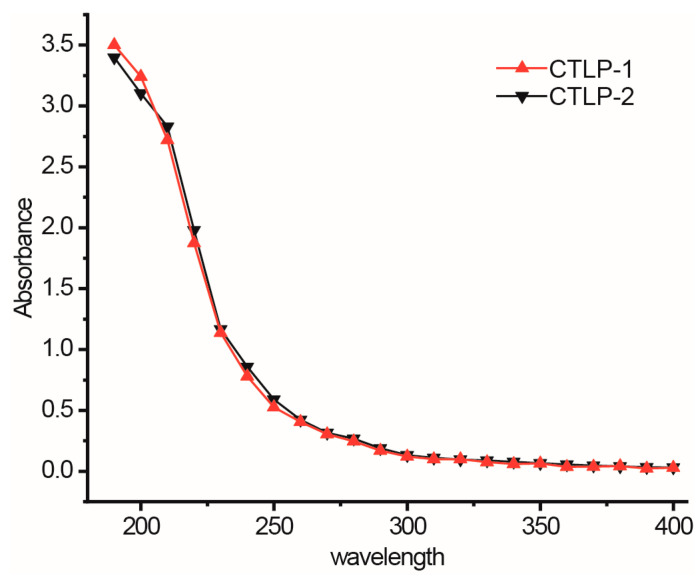
The UV spectrum of CTLP-1 and CTLP-2.

**Figure 3 polymers-14-03510-f003:**
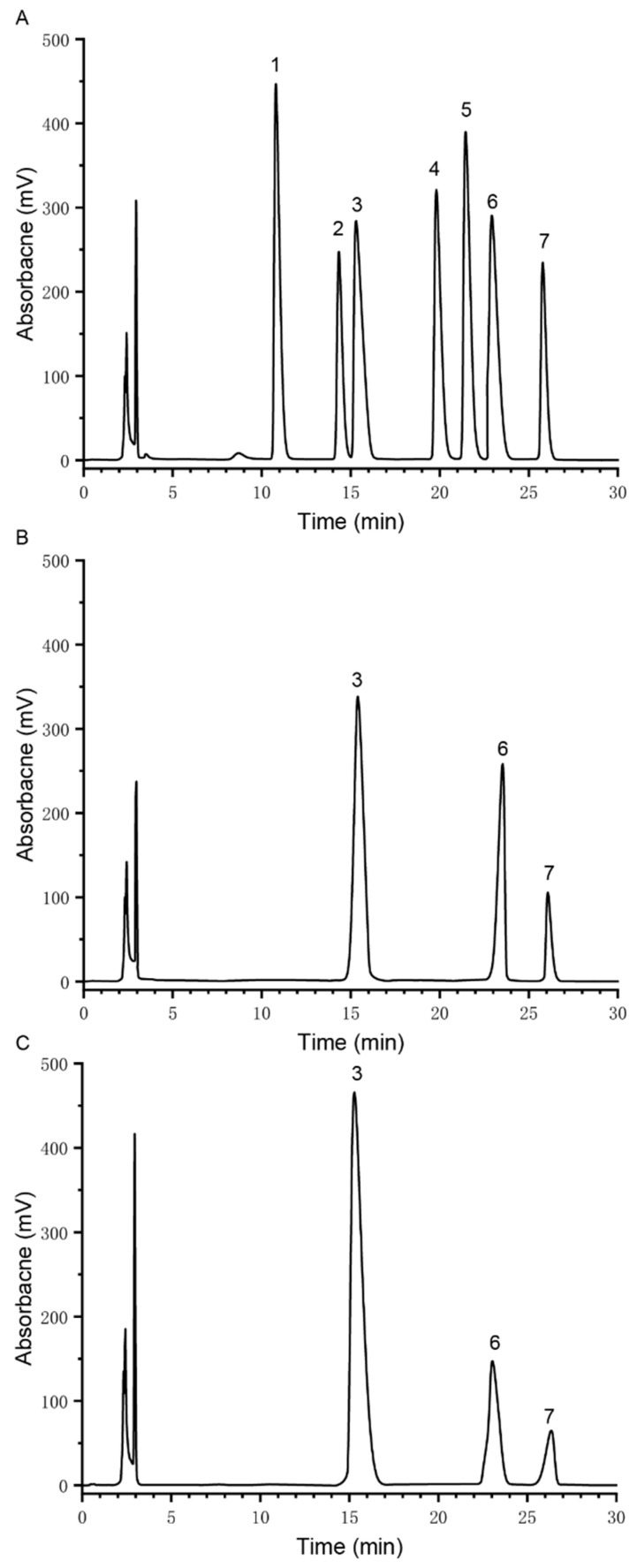
HPLC chromatographs of the monosaccharide standard (**A**), CTLP-1 (**B**), and CTLP-2 (**C**): 1: fructose; 2: rhamnose; 3: arabinose; 4: xylose; 5: mannose; 6: glucose; 7: galactose.

**Figure 4 polymers-14-03510-f004:**
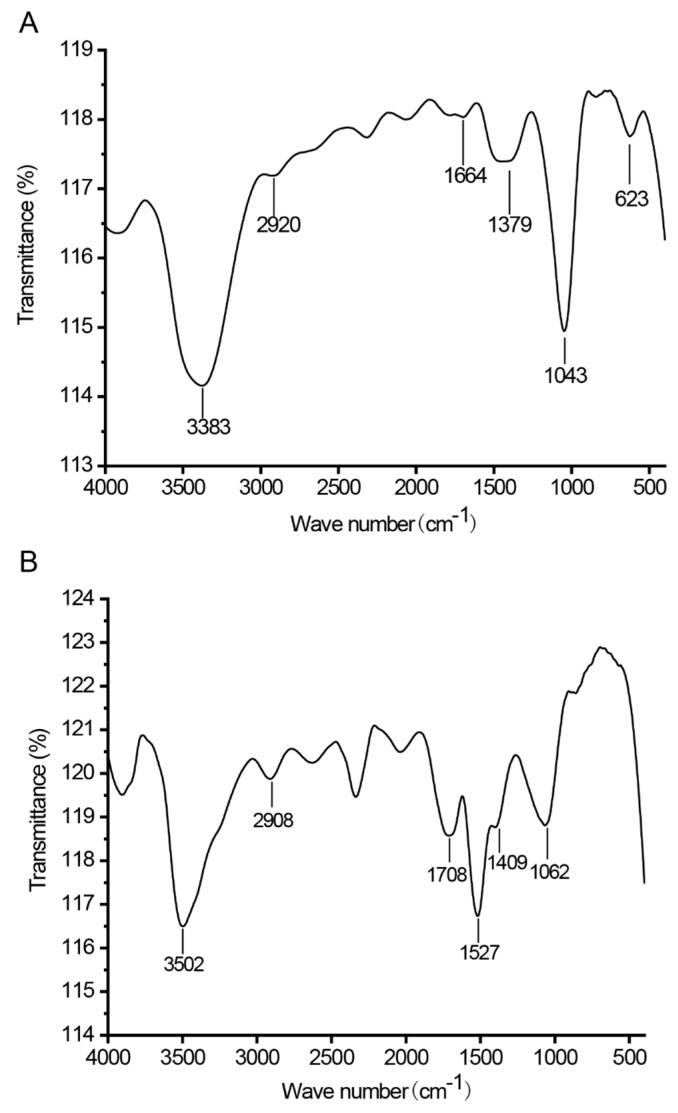
FT-IR spectra of CTLP-1 (**A**) and CTLP-2 (**B**).

**Figure 5 polymers-14-03510-f005:**
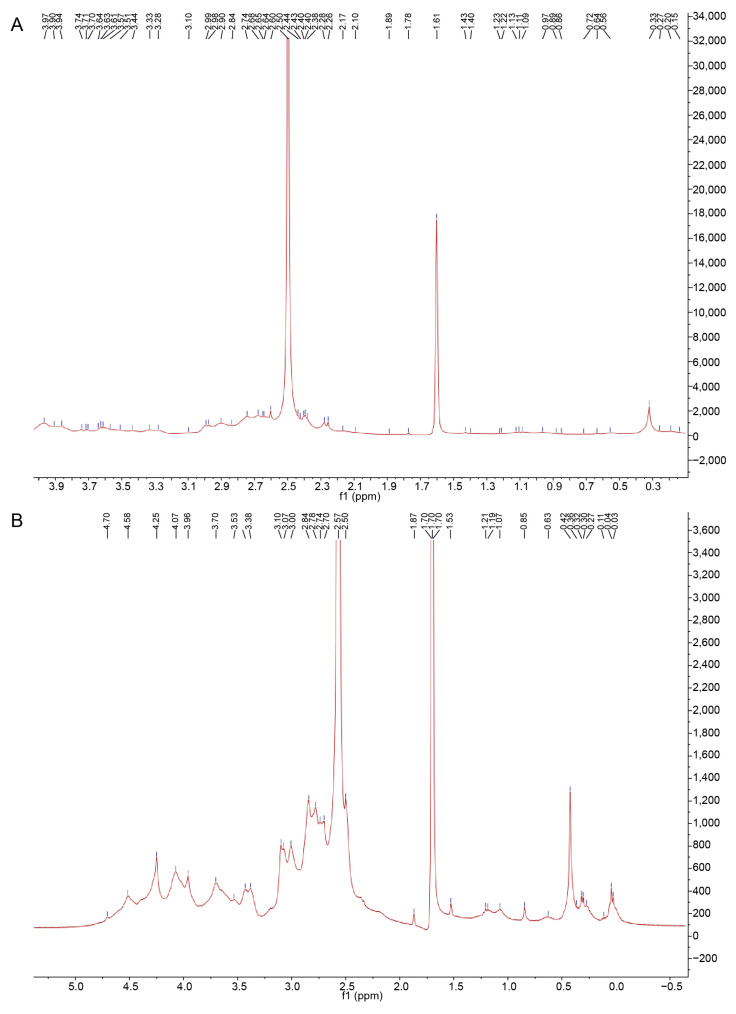
^1^H-NMR spectra of CTLP-1 (**A**) and CTLP-2 (**B**).

**Figure 6 polymers-14-03510-f006:**
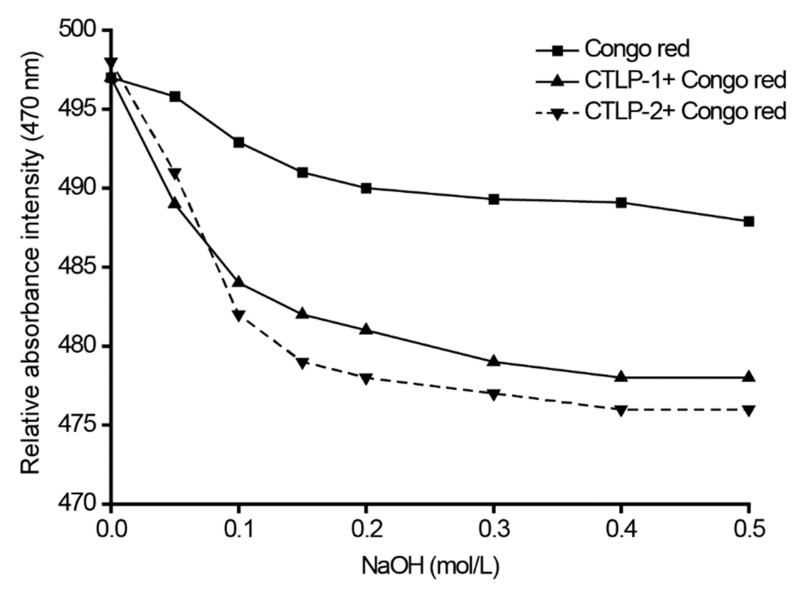
Congo red analysis results for CTLP-1 and CTLP-2.

**Figure 7 polymers-14-03510-f007:**
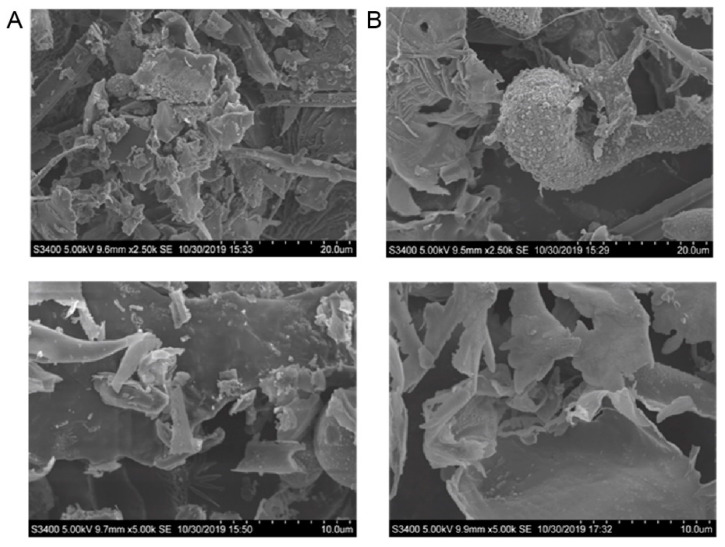
SEM images of CTLP-1 (**A**) and CTLP-2 (**B**) at 20 nm and 10 nm.

**Figure 8 polymers-14-03510-f008:**
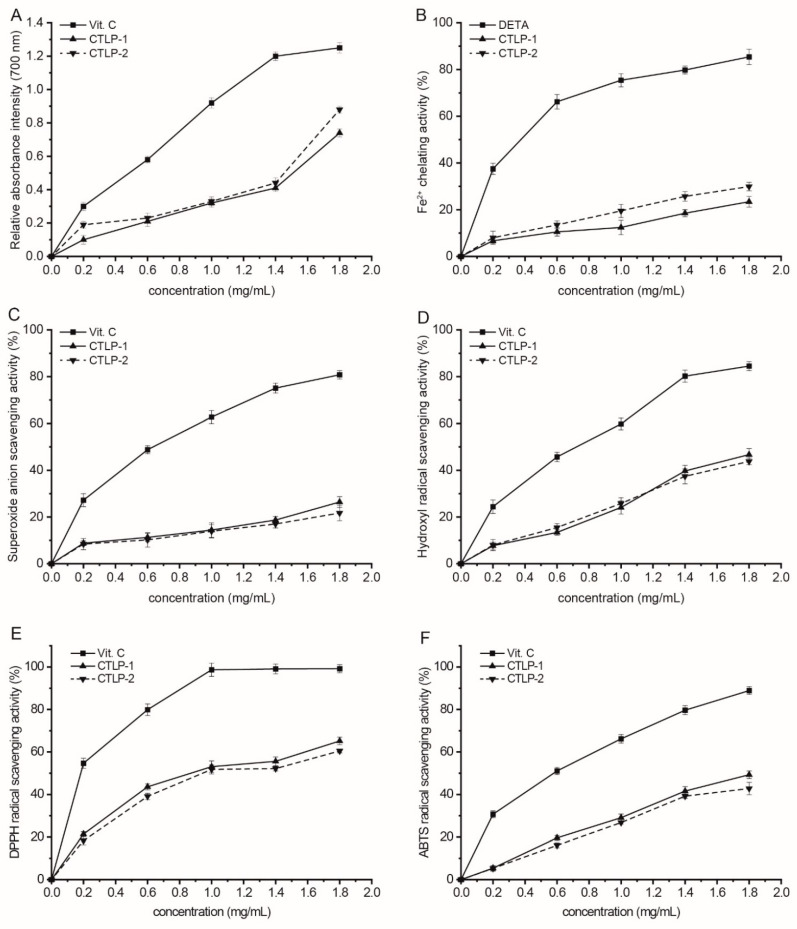
Antioxidant activities of CTLP-1 and CTLP-2. (**A**) reducing power; (**B**) Fe^2+^ chelating activity; (**C**) superoxide anion scavenging activity; (**D**) hydroxyl radical scavenging activity; (**E**) DPPH radical scavenging activity; and (**F**) ABTS radical scavenging activity.

## Data Availability

Not applicable.

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
