# Peer review of "Chemical Structures and Antioxidant Activities of Polysaccharides from *Carthamus tinctorius* L."

_polymers, 2022, doi:10.3390/polym14173510_

Round 1
Reviewer 1 Report
Though the subject of the proposed manuscript looks interesting and on the first glance a decent battery of methods was used, I can not evaluate the manuscript due to, at very least, strange use of scientific sources.
In the Materials and Methods section (l.104-166), the methodology description is inconsistent. Ref. 16 doesn't describe neither periodate, nor FRAP or OH⦁ scavenging methods; ref. 17 doesn't describe DPPH⦁ method; ref. 21 doesn't describe neither ABTS⦁+, nor O2⦁- scavenging methods.
In Results and Discussion section (l.220), in the sentence "...The periodate consumption and formic acid production differ based on the reaction conditions [21,22]...", two references are cited. Neither ref. 21, nor ref. 22 comment on periodate reaction, in [21] it was not used at all, in [22] - only for silver staining of gels.
I assumed that no further reading of the manuscript and examination of references is justified, as well as enumeration of the style/presentation recommendations I prepared before founding of these unwarranted citations.
Author Response
Dear Reviewer:
We are very grateful to you for reviewing the paper so carefully. We have carefully considered your suggestion and made some changes.
In the Materials and Methods section (l.104-166), the methodology description is inconsistent. Ref. 16 doesn't describe neither periodate, nor FRAP or OH⦁ scavenging methods; ref. 17 doesn't describe DPPH⦁ method; ref. 21 doesn't describe neither ABTS⦁+, nor O2⦁- scavenging methods.
In Results and Discussion section (l.220), in the sentence "...The periodate consumption and formic acid production differ based on the reaction conditions [21,22]...", two references are cited. Neither ref. 21, nor ref. 22 comment on periodate reaction, in [21] it was not used at all, in [22] - only for silver staining of gels.
Response: Your suggestions and criticisms are very reasonable and accurate. We are very sorry for our inappropriate and incorrect references. This is really a severe mistake, which revealed our carelessness in writing the manuscript and our lax review before submission. We have reexamined the entire manuscript and revised and updated the references.
Including but not limited to: updated ref. 16, 21, 22, 23, 24 and 25 in line 110-170, updated ref. 16 in line 232. Among them, ref. 16 described periodate oxidation, ref. 21 described the determination reducing power, ref. 22 described FRAP and DPPH methods, ref. 23 described the superoxide anion radical scavenging method, ref. 24 described the hydroxyl radical scavenging method, ref. 25 described ABST method.
We are sorry again for our carelessness. We sincerely hope that this revised manuscript has addressed all your comments and suggestions. We appreciated the reviewers’ warm work earnestly, and hope that the correction will meet with approval. Once again, thank you very much for your comments and suggestions.
Reviewer 2 Report
The authors of the manuscript “Chemical Structures and Antioxidant Activities of Polysaccharides from Carthamus tinctorius L.” present an interesting work on two polysaccharides from C. tinctorius L. (CTLP-1 and CTLP-2). These polysaccharides were purified, and their structures were analyzed by physical and chemical testing. Their antioxidant activty have been also tested.
The manuscript is clearly written and the results are interesting but contains some flaws that need to be improved before acceptance.
Line 11: authors wrote “Carthamus tinctorius L.”, please use italic format.
Line 41: the reference [13] is not available or i was not able to recover it. Please, may authors add some lines for better explain the story of CTLPs.
Lines 86 and 179: please, may authors describe the "DNS Method"? I didn’t find any 3,5-Dinitrosalicylic acid method.
Lines 177-178: Why did authors use a discontinous gradient for eluiting CTLPs?
Line 180: authors shouls specify the dialysis condition (buffer, membrane cut off or other matherial) on chapter 2.
Line 189: Figure S1 can be inserted as new Fig 2. Remember, this shift the numbers of other figures.
Line 194: The figure S2 can be omitted, the equation is enough.
Lines 197–202: authors describes an hazardous relationship between size of sugars and their biological activity. Please, may authors report/describes at least few examples?
Lines 208-209: Please write somewhere “Fig 2B and Fig 2C”.
Line 222: The figure S3 can be omitted, the equation is enough. You do not need anymore “supplementary materials file”.
Line 261: Authors may use “Panel” A and B instead “upper” and “under”
Line 273: Figure 5 – please change the tick marks on CTLP curves, the “ordinate” should be “relative absorbance intensity”
Line 281: Figure 6 – please, use “Panel” A …. D instead of “upper” and “under”, probably some other information from SEM is necessary.
Lines 282-284: Please, delete these lines.
Line 295: Figure 7 - please change the tick marks on CTLP/Vitamine C curves, they are too similar. Use also dotted line etc…
Lines 323-326: Please, maybe it is necessary explain very well on introduction “why is it necessary to perform antioxidant activity?”
Line 329: Please, change “galactos.” with “galactose.”
Author Response
Dear Reviewer:
We are very grateful to you for reviewing the paper so carefully. We have carefully considered your suggestion and made some changes.
Line 11: authors wrote “Carthamus tinctorius L.”, please use italic format.
Response: We are very sorry for our incorrect writing and it was rectified in line 11.
Line 41: the reference [13] is not available or i was not able to recover it. Please, may authors add some lines for better explain the story of CTLPs.
Response: We are very sorry for our negligence of the explanation. Reference [13] is our previous work. First we improved the extraction rate of CTLPs by ultrasonic assisted extraction method. Then in this manuscript, its chemical structures and antioxidant activity in vitro were studied. At present, we are measuring antioxidant the activity of CTLPs in vivo, and researching the relationship between its activity and structure, and even we are trying to explore the molecular mechanism of its antioxidant effect.
The relevant results (reference [13]) have been published in Basic & Clinical Pharmacology & Toxicology and which can be retrieved in the form of abstract, with links (Page 253).
Abstracts of the 2020 International Conference on Electronic Healthcare Technologies, 26 June 2020, Toronto, Canada: Basic & Clinical Pharmacology & Toxicology: Vol 127, No S1 (wiley.com)
https://onlinelibrary.wiley.com/action/doSearch?AllField=Ultrasonic-Assisted+Extraction+Optimization%2C+Structural+Elucidation+and+Antioxidant+Activity+of+Polysaccharides+from+Carthamus+tinctorius+L&SeriesKey=17427843
Lines 86 and 179: please, may authors describe the "DNS Method"? I didn’t find any 3,5-Dinitrosalicylic acid method.
Response: We are very sorry for our negligence of the explanation. In brief, DNS, i.e. dinitrosalicylic acid method, is based on the principle that dinitrosalicylic acid (DNS) reacts with reducing sugar under alkaline conditions to generate 3-amino-5-nitrosalicylic acid. The product is brownish red under boiling conditions, and the color depth is proportional to the content of reducing sugar within a certain concentration range. The content of reducing sugar is determined by colorimetric method.
The first is our mistake. The DNS method should not appear in line 86, we have erased it. Second, in line 85, we cited the literature[14] on the DNS method as follow.
[14] Saqib, A.A.N.; Whitney, P.J. Differential behaviour of the dinitrosalicylic acid (DNS) reagent towards mono- and di-saccharide sugars. Biomass Bioenergy 2011, 35, 4748-4750.
Lines 177-178: Why did authors use a discontinous gradient for eluiting CTLPs?
Response: We thank the reviewer for pointing this out. In our previous pre experiment, two different components, CTLP11 and CTLP2 were obtained after gradient elution of CTLPs. Subsequently, CTLP11 and CTLP2 were eluted by gradient, and only a single component was obtained, indicating that the purity of CTLP11 and CTLP2. Therefore, in the subsequent experiments, in order to prepare CTLP11 and CTLP2 in large quantities, we directly eluted them with 0.1 mol∙L−1 and 0.2 mol∙L−1 NaCl solution respectively.

Reviewer 3 Report
General comment:
The manuscript “Chemical Structures and Antioxidant Activities of Polysaccharides from Carthamus tinctorius L.” report the purification and characterization of two polysaccharides, namely CTLP-1 and CTLP-2) from the Safflower. Several tools (Absorbance, HPLC, FTIR, NMR..) were employed to confirm the purity and the identification of purified compounds. The evaluation of the antioxidant effect was accomplished using several radicals, which tests the activity of different functional groups and makes the conclusion more wholistic. The use of approaches from different nature for the identification, the characterization and functional properties makes the work complete and suitable for publication, after considering minor revision.
Precise comment to consider:
- The equation and the corresponding text are not clear. Please check the reference again and correct. (Lines from 167 to 171 in page 4).
- Line 282 page 11, There a title named Abstract. Please correct. The text included here seems to be out of context. Please correct.
- The English of the last part of results and discussion should be improved. Repeating the sentence “As shown in fig7” was not professional and affect the aesthetic of the manuscript.
- The conclusion is mostly a repetition of the Abstract. Please improve by stating for example the soundness of the work and possible future/perspective ideas.
Author Response
Dear Reviewer:
We are very grateful to you for reviewing the paper so carefully. We have carefully considered your suggestion and made some changes.
The equation and the corresponding text are not clear. Please check the reference again and correct. (Lines from 167 to 171 in page 4).
Response: We are very sorry for our incorrect writing. We have modified the equation and the corresponding text in line 175 which is helpful for identification.
Line 282 page 11, There a title named Abstract. Please correct. The text included here seems to be out of context. Please correct.
Response: Thank you very much for your careful review. We are very sorry for our carelessness and the lines 282-284 would be deleted.
The English of the last part of results and discussion should be improved. Repeating the sentence “As shown in fig7” was not professional and affect the aesthetic of the manuscript.
Response: We are very sorry for our incorrect writing and we have done it according to your suggestion. We have revised the relevant content to make the language of our manuscript more professional in line 296-332.
The conclusion is mostly a repetition of the Abstract. Please improve by stating for example the soundness of the work and possible future/perspective ideas.
The two CTLP components were obtained and marked as CTLP-1 and CTLP-2 by using a G-100 agarose gel column from Carthamus tinctorius L. The relative molecular weights of CTLP-1 and CTLP-2 were 5900 and 8200 Da, respectively. CTLP-1 was a heteropolysaccharide composed of arabinose, glucose, and galactose with a molar ratio of 6.7: 4.2: 1, CTLP-2 was a heteropolysaccharide containing arabinose, glucose, and galactose with a molar ratio of 16.76: 4.28: 1. The results showed that both CTLP-1 and CTLP-2 exhibit high antioxidant activity in vitro and could be further applied as a novel natural antioxidant drug. In future we are measuring the activity of CTLPs in vivo, and researching the relationship between its activity and structure.
We sincerely hope that this revised manuscript has addressed all your comments and suggestions. We appreciated the reviewers’ warm work earnestly, and hope that the correction will meet with approval. Once again, thank you very much for your comments and suggestions.
Reviewer 4 Report
Dear Authors,
the manuscript entitled "Chemical structures and antioxidant activity of polysaccharides of Carthamus tinctorius L." concerns the study of two purified polysaccharides from Carthamus tinctorius and their antioxidant activities. In the literature, the two polysaccharides have already been purified and the data have been published in a paper by Hu entitled "Structural analysis and antioxidant activity of polysaccharide isolated from Carthamus tinctorius L. [dissertation]. Shihezi: Shihezi University", however, the antioxidant assays used in this manuscript were used for the first time for the purpose of evaluating antioxidant activities, and thus with this paper the authors contribute to complete the framework of the aforementioned activities. However, minor corrections are necessary for publication in Polymers.
Abstract:
page 1, line 11: Carthamus tinctorius should be written in italics
Introduction:
Page 2,line 51: briefly insert the results obtained
Results:
For antioxidant activities I would also calculate EC50 and IC50 values as appropriate.
Page 11, line 282-284: delete
References:
Double-check all references and write all plant and fungi names in italics
Author Response
Dear Reviewer:
We are very grateful to you for reviewing the paper so carefully. We have carefully considered your suggestion and made some changes.
Abstract:
page 1, line 11: Carthamus tinctorius should be written in italics
Response: We are very sorry for our incorrect writing and it was rectified in line 11.
Introduction:
Page 2,line 51: briefly insert the results obtained
Response: We thank the reviewer for pointing this out. We agree and have updated in line 55 as follows.
The molecular weight, monosaccharide composition and main sugar chain skeleton of CTLP were determined. And CTLP exhibited high reducing power, hydroxyl radical scavenging activity, DPPH radical scavenging activity and ABTS radical scavenging activity, moderate Fe2+ chelating activity and superoxide anion scavenging activity.
Results:
For antioxidant activities I would also calculate EC50 and IC50 values as appropriate.
Response: We appreciate it very much for this good suggestion. In fact, in China, safflower is commonly used as a food and Chinese herbal medicine in daily life, and its safety had been proven. In this manuscript, we investigated the structure and in vitro antioxidant activity of CTLP. Meanwhile, we are measuring the activity of CTLPs in vivo, and researching the relationship between its activity and structure. And even we are trying to explore the molecular mechanism of its antioxidant effect.
Page 11, line 282-284: delete
Response: Thank you very much for your careful review. We are very sorry for our carelessness and the lines 282-284 had been deleted.
References:
Double-check all references and write all plant and fungi names in italics
Response: We are very sorry for our incorrect writing. According to your suggestion, all the relevant names of plant and fungi in the manuscript have been changed to italics.
We sincerely hope that this revised manuscript has addressed all your comments and suggestions. We appreciated the reviewers’ warm work earnestly, and hope that the correction will meet with approval. Once again, thank you very much for your comments and suggestions.

Round 2
Reviewer 1 Report
Upon corrections made in methodological references (which now look good), it became possible to review the manuscript as such.
There are some style mistakes/typos which are left from the 1st version:
Abstract:
l.13 ...CTLP-1 had a mass of 5900 Da which was composed... (compare with ...CTLP-2 had a mass of 8200 Da and was composed...)
1. Introduction
l.30 ...is good fo improving...
l.36 ...safflow yellow... is this misspelled "safflower yellow"? If yes, carthamin and safflow yellow are synonyms
l.42 ...our laboratory optimized the extraction conditions of polysaccharides from C. tinctorius L. (CTLP), and high CTLP yield has been obtained in published articles[13]. - well, high yield is obtained in the experiments described in published article.
2. Materials and Methods
l.62 C. tinctorius L. was acquired from Tacheng County (Xinjiang, China) - usual practices require mentioning of the plant sample depositors, if the plant was collected, not purchased. What means "acquired" in this context?
l.68 ...DEAE-52 cellulose column material was pretreated, and the activated column... - how was DEAE-52 pretreated (for activation, I suppose)?
l.99-101, l.113-115 - is it necessary to describe drawing of the standard curve?
l.105, l.107 ...(mobile phase A) was passed through the organic membrane... what was the porosity of the membrane?
3. Results and discussion
l.209 ...root polysaccharide of Panax ginseng marc was more complex... - is the systematic name spelled correctly?
l.227 - However, these trends are not seen in all studies, and the specific relationships remained remain to be determined.
l.305 - Fig 8. - abbreviations of the reference compounds are not conventional - maybe vit.C would be better than Vc, DETA-2Na should be corrected to EDTA.
l.330 - ...We also noted that the ATBS radical scavenging activities of CTLP-1 and CTLP-2 increased with increasing polysaccharide concentration... - was something else expected?
4. Conclusion
l.344-345 - ...The results showed that both CTLP-1 and CTLP-2 exhibit high antioxidant activity in vitro and could be further applied as a novel natural antioxidant drug... - This conclusion is not founded by the authors' observations: fig.8 shows that all reported AO capacities are much lower than those of the referent compound (vit.C or EDTA); moreover, authors note this fact - l. 310, 315,
Author Response
Dear Reviewer:
We sincerely thank you for thoroughly examining our manuscript and providing very helpful comments to guide our revision. We have carefully considered your suggestion and made some changes.
Abstract:
l.13 ...CTLP-1 had a mass of 5900 Da which was composed... (compare with ...CTLP-2 had a mass of 8200 Da and was composed...)
Response: We thank the reviewer for pointing this out. We have updated in lines 16 as follows: CTLP-2 had a mass of 8200 Da which was composed of arabinose, glucose, and galactose…
- Introduction
l.30 ...is good fo improving...
Response: We are very sorry for our incorrect writing and it was rectified in line 30.
l.36 ...safflow yellow... is this misspelled "safflower yellow"? If yes, carthamin and safflow yellow are synonyms
Response: We are very grateful to you for reviewing the paper so carefully. We rechecked the meaning of "safflow yellow" and "safflower yellow". They had appeared in different literatures and were components of safflower which they should be synonyms. After our consideration, "safflower yellow" should be more appropriate in this manuscript and we have updated in line 36.
l.42 ...our laboratory optimized the extraction conditions of polysaccharides from C. tinctorius L. (CTLP), and high CTLP yield has been obtained in published articles[13]. - well, high yield is obtained in the experiments described in published article.
Response: We are very sorry for our incorrect writing and which was rectified in line 41, 42.
- Materials and Methods
l.62 C. tinctorius L. was acquired from Tacheng County (Xinjiang, China) - usual practices require mentioning of the plant sample depositors, if the plant was collected, not purchased. What means "acquired" in this context?
Response: We are very sorry for our negligence of the explanation and we have updated in line 62. C. tinctorius L. was kindly provided by Xinjiang Academy of Agricultural Sciences (Xinjiang, China) which was purchased from local herdsmen in Tacheng, Xinjiang, China.
l.68 ...DEAE-52 cellulose column material was pretreated, and the activated column... - how was DEAE-52 pretreated (for activation, I suppose)?
Response: We thank the reviewer for pointing this out. The purpose of pretreatment is to activate DEAE-52. Most column materials should be activated before use. The activation method of DEAE-52 in this manuscript is as follows: DEAE-52 was weighed and soaked in distilled water overnight, during which water was changed several times and fine particles were removed each time. Then soaked it for more than 1h in 0.5M NaOH solution, drained it and rinsed it with deionized water to make the pH about 8. Then soaked the solution in 0.5M HCl solution for more than 1h, removed the acid solution, and washed it with deionized water to about pH 6. Soak in 0.0175M, pH 6.7 phosphate buffer and equilibrate for reserve use.
l.99-101, l.113-115 - is it necessary to describe drawing of the standard curve?
Response: We appreciate it very much for your good suggestion. These are simple and common methods for experts in relevant professional fields like you. Nevertheless, some reviewers suggested that the method part be clearly described. Therefore, the detailed description of some methods may not be deleted. We sincerely ask for your understanding on this point.
l.105, l.107 ...(mobile phase A) was passed through the organic membrane... what was the porosity of the membrane?
Response: We are very sorry for our negligence of the tagging and we have marked in line 107. Pure acetonitrile (mobile phase A) was passed through the organic membrane (0.22 μm).
- Results and discussion
l.209 ...root polysaccharide of Panax ginseng marc was more complex... - is the systematic name spelled correctly?
Response: We thank the reviewer for pointing this out. The systematic name "Panax ginseng marc" should be correctly, which was from ref. 18 as follows:
[18] Li, J.; Wang, D.; Xing, X.; Cheng, T.-J.R.; Liang, P.-H.; Bulone, V.; Park, J.H.; Hsieh, Y.S.Y. Structural analysis and biological activity of cell wall polysaccharides extracted from Panax ginseng marc. Int. J. Biol. Macromol. 2019, 135, 29-37.
l.227 - However, these trends are not seen in all studies, and the specific relationships remained remain to be determined.
Response: We are very sorry for our carelessness and incorrect writing and which was rectified in line 228.
l.305 - Fig 8. - abbreviations of the reference compounds are not conventional - maybe vit.C would be better than Vc, DETA-2Na should be corrected to EDTA.
Response: We are very grateful to you for reviewing the paper so carefully. We think your suggestion is correct and necessary. We have modified and updated the abbreviations in the manuscript, including the Fig 8.
l.330 - ...We also noted that the ATBS radical scavenging activities of CTLP-1 and CTLP-2 increased with increasing polysaccharide concentration... - was something else expected?
Response: We thank the reviewer for pointing this out. In this manuscript, we investigated the structure and in vitro antioxidant activity of CTLP. Meanwhile, we are measuring the activity of CTLPs in vivo, and researching the relationship between its activity and structure. And even we are trying to explore the molecular mechanism of its antioxidant effect.
- Conclusion
l.344-345 - ...The results showed that both CTLP-1 and CTLP-2 exhibit high antioxidant activity in vitro and could be further applied as a novel natural antioxidant drug... - This conclusion is not founded by the authors' observations: fig.8 shows that all reported AO capacities are much lower than those of the referent compound (vit.C or EDTA); moreover, authors note this fact - l. 310, 315,
Response: We appreciate your pertinent and reasonable suggestions. It is known that Vit. C has extremely strong reducibility, and EDTA is also a strong chelating agent. Therefore, as shown in Fig. 8, Vit. C and EDTA were used as positive controls, it was understandable that the activities of CTLP-1 and CTLP-2 were not as high as those of Vit. C and EDTA. And compared with our published research on Pleurotus ferulae polysaccharides (ref. 19) and other fungal polysaccharides we are studying, CTLP-1 and CTLP-2 exhibited higher antioxidant activities. Nevertheless, as you suggested, this conclusion is indeed not rigorous enough and we have made the following modifications in line 345: The results showed that both CTLP-1 and CTLP-2 exhibited relatively high antioxidant activity in vitro…
We are sorry again for our carelessness. We sincerely hope that this revised manuscript has addressed all your comments and suggestions. We appreciated the reviewers’ warm work earnestly, and hope that the correction will meet with approval. Once again, thank you very much for your comments and suggestions.
